# HOW TO MERGE MULTIMODAL MODELS OVER TIME?

**Sebastian Dziadzio**[*1]    **Vishaal Udandarao**[*1,2]    **Karsten Roth**[*1,3]    **Ameya Prabhu**[∘1]
**Zeynep Akata**[3†]    **Samuel Albanie**[†]    **Matthias Bethge**[1†]

[1]Tübingen AI Center, University of Tübingen  [2]University of Cambridge
[3]Munich Center for ML, Technical University of Munich

## ABSTRACT

Model merging combines multiple "expert" models finetuned from a base foundation model on diverse tasks and domains into a single, more capable model. However, most existing model merging approaches assume that all experts are available simultaneously. In reality, new tasks and domains emerge over time, requiring strategies to integrate the knowledge of expert models as they become available: a process we call *temporal model merging*. The temporal dimension introduces unique challenges not addressed in prior work, raising new questions such as: when training for a new task, should the expert model start from the merged past experts or from the original base model? Should we merge all models at each time step? Which merging techniques are best suited for temporal merging? Should different strategies be used to initialize the training and deploy the model? To answer these questions, we propose a unified framework called TIME (Temporal Integration of Model Expertise) which defines temporal model merging across three axes: (1) initialization, (2) deployment, and (3) merging technique. Using TIME, we study temporal model merging across model sizes, compute budgets, and learning horizons on the FoMo-in-Flux benchmark. Our comprehensive suite of experiments across TIME allows us to build a better understanding of current challenges and best practices for effective temporal model merging.

## 1    INTRODUCTION

Foundation models consolidate a wide range of capabilities and knowledge into a single, large model. Consequently, model merging (Regent's et al., 1996; Wortsman et al., 2022a) has emerged as a key technique for unifying multiple task-specific specialist models derived from a shared base into a single, generalist model.

Current model merging approaches typically assume a fixed base model that is fine-tuned independently on $k$ diverse tasks and domains to produce a set of *independent* experts (Garipov et al., 2018; Rofin et al., 2022; Ilharco et al., 2022; Yadav et al., 2023; Li et al., 2022), which are then merged simultaneously. Research in this field has therefore focused on improving merging techniques for larger or structurally different $k$-sets, exploring the impact of the diversity and scale of finetuning domains, tasks and experts.

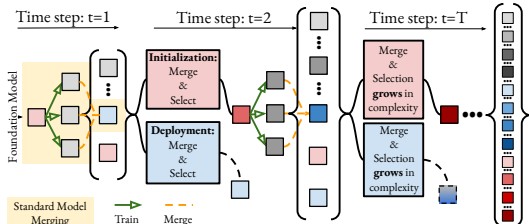

Figure 1: **Temporal Model Merging** generalizes standard model merging (yellow), which merges multiple trained experts once, in a single step. Our study reveals that initialization and deployment strategies are crucial in the temporal setting.

However, the world is constantly evolving, leading to shifts over data distributions, domains, and tasks, with new concepts emerging that may been insufficiently covered during large-scale pretraining (Koh et al., 2021; Hu et al., 2022; Pratt et al., 2023; Menon & Vondrick, 2023; Zhang et al., 2021; Gui et al., 2024). This dynamic nature of real-world applications motivates a hitherto missing systematic exploration into *temporal model merging* (see Fig. 1) to better understand model

---

[*]equal contribution, random order †equal supervision, random order,∘ core contributor.
Correspondence to: {vishaal.udandarao, sebastian.dziadzio}@bethgelab.org

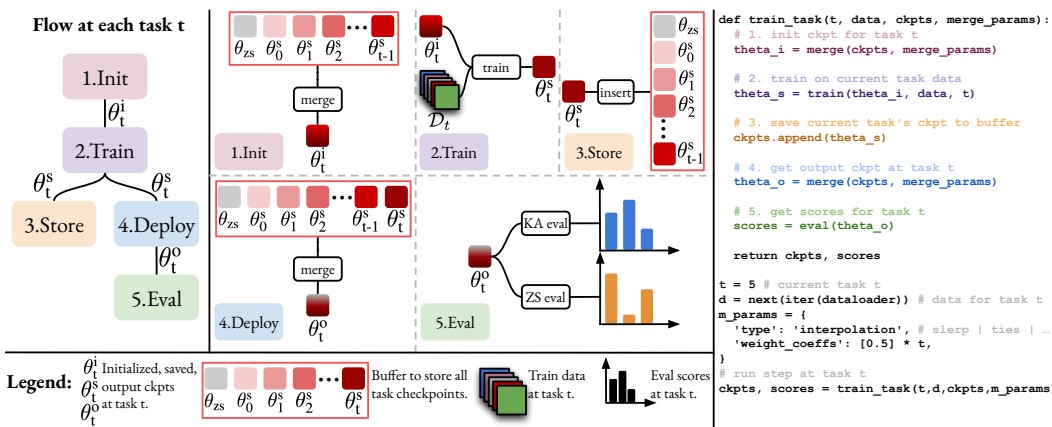

Figure 2: **Design Space of Temporal Model Merging through TIME.** We showcase our framework for the per-task pipeline of temporal model merging over multiple tasks: At each task $t$, we first initialize the current checkpoint to start training from, $\theta_t^i$, by using one or more previously stored checkpoints from previous tasks, either directly or by merging them. We train $\theta_t^i$ on current task data $\mathcal{D}_t$ to yield the current task checkpoint $\theta_t^s$, which is inserted into the checkpoint buffer. Finally, to produce the output model, $\theta_t^o$, we either merge previously stored checkpoints from the buffer or use them directly. The entire framework is depicted in the pseudo-code on the right panel.

merging along an additional, understudied axis: *time* (Zhou et al., 2024; Don-Yehiya et al., 2022). Specifically, in this work, we ask: **(1)** What is the best merging strategy to initialize each expert model before training? **(2)** What is the best merging strategy to deploy each expert model after training? **(3)** Which model merging techniques are most suitable for temporal merging? To answer these questions, we propose a unified framework for studying temporal model merging: TIME (Temporal Integration of Model Expertise). It is structured around three key axes spanning the design space of temporal merging solutions (as shown in Fig. 2): the choice of past checkpoints to merge before training on the current task (initialization), the choice of past checkpoints to merge after training on the current task (deployment), and the choice of the merging technique. Using our TIME framework, we position existing model merging approaches along each key axis and conduct a systematic study of model merging over time. For this, we leverage the multimodal FoMo-in-Flux by Roth et al. (2024b), a benchmark comprising 63 datasets with well-documented sequential properties, enabling a thorough investigation of temporal model merging under practical compute constraints. Our experiments systematically explore different merging techniques, initialization, and deployment strategies, providing several key insights:

> **Key Insights for Temporal Model Merging**
>
> **[A] Accounting for time is crucial.** Standard "*offline*" model merging techniques do not generalize well to the temporal merging setting (Sec. 3.1).
>
> **[B] Complex merging techniques matter little.** Choosing sophisticated merging techniques beyond simple weighted averaging provides at best marginal benefits for temporal model merging, especially for long task sequences (Sec. 3.3).
>
> **[C] Initialization and deployment are critical.** Choosing how to select and combine available weights before and after each task $t$ is most important for temporal model merging (Sec. 3.2).
>
> **[D] Temporal model merging scales well.** Larger models or compute budgets allows greater benefits from temporal merging. Scaling enables temporal model merging to even outperform the multitask model, trained on all tasks at once (Sec. 3.4).

## 2  DESIGN SPACE OF TEMPORAL MODEL MERGING

**Notation.** Throughout this work, we use $t$ to refer to a given task at time $t$. Full model parameterization is denoted by $\theta$, with the following key instantiations: $\theta_t$ represents model weights at task $t$, while $\theta_t^I$, $\theta_t^S$, and $\theta_t^O$ denote weights used for *initialization*, *saved* weight checkpoints at task $t$ (i.e. the trained expert models), and the *output* deployed model weights, respectively. Note that

Table 1: **Comparison of model merging techniques.**

| Method | Sparsification | Consensus | Scaling |
|---|---|---|---|
| Weight averaging Regent's et al. (1996); Wortsman et al. (2022a) | ✗ | Linear Int. | Weight coeff. |
| SLERP Ramé et al. (2024) | ✗ | Spherical Int. | Weight coeff. |
| Task Arithmetic Ilharco et al. (2023) | ✗ | Linear Int. | Scaling factor |
| MagMax Marczak et al. (2024) | ✗ | Max. Magnitude | Scaling factor |
| TIES Yadav et al. (2023) | Top-k | Sign Agreement | Scaling factor |
| DARE-TIES Yu et al. (2024) | Random | Sign Agreement | Scaling factor |
| Breadcrumbs-TIES Davari & Belilovsky (2025) | Top/Bottom-k | Sign Agreement | Scaling factor |
| Model Stock Jang et al. (2024) | ✗ | Geometric | Adaptive ratio |
| LiNeS Wang et al. (2024a) | ✗ | ✗ | Layer weights |

while standard model merging considers model weights as elements of a fixed set $\{\theta_k\}_{k=1}^K$, temporal model merging organizes them along the time axis $\theta_t$.

## 2.1 Temporal Model Merging through TIME

Standard model merging is typically performed offline, after all experts have been trained to convergence. In contrast, model merging in continual pretraining is generally done sequentially, using past checkpoints. Both approaches are specific instances of our more general temporal merging framework, TIME, which defines temporal merging along *three key axes*: initialization of each expert, merging for deployment at step $t$, and merging techniques $f_{\text{merge}}$ applied over time:

### Axis 1: Initialization

As expert models are created continuously over time, initialization becomes a crucial choice. Unlike model merging at a single point in time, the number of potential starting points grows exponentially over time as new experts are created. This raises the question: should starting points for each time step be derived from the base weights (as in traditional merging), from a merged combination of previous experts, or from most recent weights? In this work, we study the following initialization protocols at time step $t$ for TIME: **(1)** $\text{init}_{\text{ZS}}$, which consistently initializes with the base zero-shot model weights $\theta_0$ at each timestep $t$; **(2)** $\text{init}_{\text{FT}}$, which at step $t$ always initializes with the latest available finetuned model weights $\theta_{t-1}^S$; and **(3)** $\text{init}_{\text{EMA}}$, which computes an unrolled exponential moving average merge over all previously seen expert models $\{\theta_{t'}^S\}_{1,\dots,t-1}$ following the equation:

$$\theta_{t'}^{\text{EMA}} = f_{\text{merge}}\left(\theta_{t'-1}^{\text{EMA}}, \theta_{t'-1}^S, \mathcal{F}\right) \tag{1}$$

with merging hyperparameters $\mathcal{F}$. Consequently, the initialization weights are given as $\theta_t^I = \theta_t^{\text{EMA}}$.

### Axis 2: Deployment

With each update iteration and expert training phase $t$, a decision must be made on the final model to deploy, determining which weights to present for downstream use. In continual pretraining, the trained model $\theta_t^S$ is deployed directly. In contrast, standard model merging applies a merging technique $f_{\text{merge}}$ to a fixed set of $k$ expert models. Temporal model merging, however, must account for both previously deployed models and the growing number of expert models available over time. Unlike standard merging, where $k$ remains constant, the number of experts to merge increases with each step. As a result, temporal merging introduces the idea of weighted combinations, balancing recent updates with retained past knowledge to achieve adaptability and stability—both critical for effective continual learning (Kirkpatrick et al., 2017; Zenke et al., 2017). In this work, we study three strategies for model deployment: **(1)** $\text{deploy}_{\text{FT}}$, which always deploys the latest finetuned expert model at step $t$, i.e., $\theta_s^O = \theta_t^S$; **(2)** $\text{deploy}_{\text{EMA}}$, which computes an unrolled exponential moving average merge over all expert models, i.e., $\theta_t^O = \theta_{t+1}^{\text{EMA}}$ following Eq. (1); and **(3)** $\text{deploy}_{\text{ALL}}$, which applies a merging technique $f_{\text{merge}}$ over all previously computed expert models $\{\theta_{t'}^S\}_1^{t-1}$.

### Axis 3: Merging Techniques

At each point in time, for both initialization and deployment, merging technique $f_{\text{merge}}$ defines *how* to combine the available expert models and checkpoints. In this work, we study nine different variants in total, shown in Tab. 1. Please refer to Appendix B for details.

## 2.2 Complete Temporal Model Merging Pipeline

Incorporating all three axes of temporal model merging, we define a five-stage update pipeline for each task $t$ (see Fig. 2), consisting of the following steps: **(1) Init:** choose one of the aforementioned initialization protocols—$\texttt{init}_{\text{ZS}}$, $\texttt{init}_{\text{FT}}$, or $\texttt{init}_{\text{EMA}}$. This produces initialization weights $\theta_t^I$ at task $t$; **(2) Train:** given $\theta_t^I$, train on current task data $\mathcal{D}_t$ within a set compute budget to produce the expert model $\theta_t^S$; **(3) Store:** append $\theta_t^S$ to storage of expert model weights $\mathcal{S}$; **(4) Deploy:** choose a deployment protocol: $\texttt{deploy}_{\text{FT}}$, $\texttt{deploy}_{\text{EMA}}$, or $\texttt{deploy}_{\text{ALL}}$, and produce the output weights $\theta_t^O = \texttt{deploy}(\mathcal{S})$; and **(5) Eval:** the deployed $\theta_t^O$ is used for downstream applications and, in our case, extensive evaluation. Note that particular choices of $\texttt{init}$, $\texttt{deploy}$ and $f_{\text{merge}}$ correspond to existing approaches, for example $(\texttt{init}_{\text{ZS}}, \texttt{deploy}_{\text{ALL}}, f_{\text{merge}}^{\text{WA}})$ simply recovers offline merging through weight averaging over experts models derived from original base weights $\theta_0$ for each task $t$. Similarly, $(\texttt{init}_{\text{FT}}, \texttt{deploy}_{\text{EMA}}, f_{\text{merge}}^{\text{WA}})$ recovers exponential moving average approaches as done in Stojanovski et al. (2022); Roth et al. (2024b).

## 3 Experiments

**Training at task $t$.** We continuously finetune and merge a ViT-B/16 CLIP Radford et al. (2021); Cherti et al. (2022) model using the standard CLIP objective. The model is pretrained on LAION-2B (Schuhmann et al., 2022). We fix the training steps for each task based on the `DataComp-Small` budget of $1.8 \times 10^9$ GFLOPS, split equally across 20 tasks. At each step, we allow unrestricted access to a pretraining data pool $\mathcal{P}$, using the same 2M random subset of `LAION-400M` as in Roth et al. (2024b). We use a cosine-decay LR schedule with a linear warmup of $10\%$, AdamW optimizer Loshchilov & Hutter (2017), a batch size of 512 and gradient norm clipping to 1. All experiments use PyTorch Paszke et al. (2019), and are run on a compute cluster using NVIDIA A100/H100s.

**Evaluation and Metrics.** We use the continual pretraining benchmark Fomo-in-Flux by Roth et al. (2024b). It includes 41 adaptation datasets and 22 separate evaluation datasets, covering visual and semantic distribution shifts. We focus on two aspects: the level of *adaptation*, reflecting performance improvement with each merging step, and *retention*, capturing the preservation of prior knowledge. Specifically, we report two metrics following Roth et al. (2024b): Knowledge Accumulation ($\mathcal{A}_{KA}$), the average accuracy (or recall@5 for retrieval) across all 41 adaptation datasets, and Zero-Shot Retention ($\mathcal{A}_{ZS}$), the zero-shot accuracy or recall@5 on all 22 held-out evaluation datasets. Additional details can be found in the supplementary.

### 3.1 Do We Need Model Merging Across Time?

The simplest approach to temporal merging is to disregard the time axis and follow the standard *offline* merging paradigm. In TIME terms, this corresponds to a configuration of $(\texttt{init}_{\text{ZS}}, \texttt{deploy}_{\text{ALL}}, f_{\text{merge}})$, which always fine-tunes the initial base weights $\theta_0$. To study the effectiveness of this strategy, we test it with various choices of $f_{\text{merge}}$ in Fig. 6, including averaging, task-arithmetic, magmax, ties, dare-ties, breadcrumbs-ties, and lines-ties. For context, we include **(1)** a simple continual fine-tuning baseline (replay), which replays on both pretraining and previous task data, **(2)** initial zero-shot ($\theta_0$) performance lower bound, and **(3)** multitask training upper bound. We visualize trajectories over time for knowledge accumulation $\mathcal{A}_{\text{KA}}$, zero-shot retention $\mathcal{A}_{\text{ZS}}$, and the geometric mean. Our results show that there are marginal differences between merging techniques when deployed in an offline manner for a temporal problem, and they all trace similar trajectories in the $\mathcal{A}_{\text{KA}}$ and $\mathcal{A}_{\text{ZS}}$ space and achieve similar final performance. Overall, however, unlike straightforward continual fine-tuning (replay), offline merging with *any* technique fails to address the temporal aspects of the problem, particularly struggling to consistently acquire new knowledge over time (as shown in Fig. 6, left).

### 3.2 TIME Travel

Since offline merging is ill suited to the temporal setting, we systematically explore the design space for temporal merging methods by testing all valid combinations of three initialization protocols and three deployment protocols described in Sec. 2.2. After discarding incompatible pairs, such as

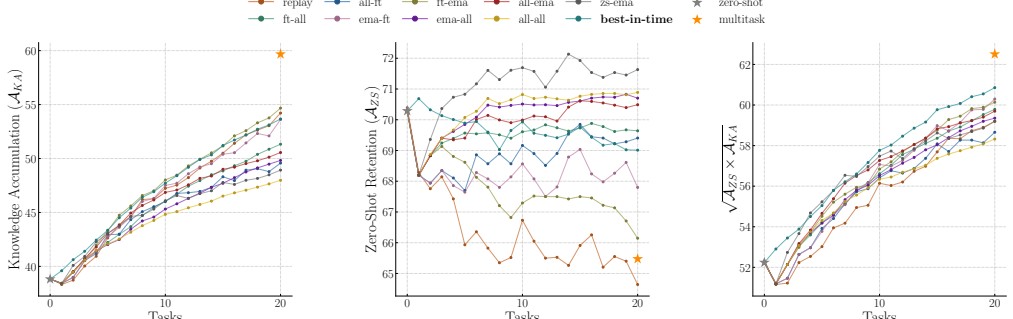

Figure 3: **A journey through TIME.** We explore various initialization and deployment protocols, finding that the EMA initialization-deployment strikes the best balance between knowledge accumulation and zero-shot retention. We refer to this strategy as *Best-in*-TIME.

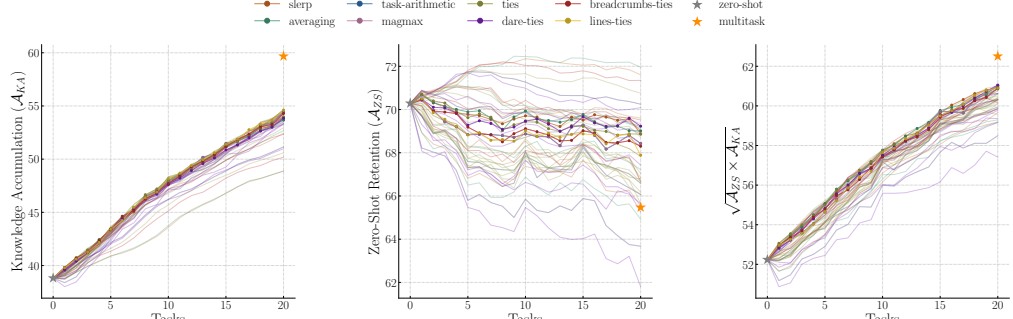

Figure 4: **Sweeping *Best-in*-TIME.** All merging techniques perform well with the *Best-in*-TIME strategy. Indeed, there are no significant differences between techniques, indicating that initialization and deployment matter more for temporal merging.

$\text{init}_{ZS}$ with $\text{deploy}_{FT}$, we evaluated the remaining eight variants using weight averaging as the merging technique. As shown in Fig. 3, the choice of initialization and deployment strategy largely determines performance, significantly affecting both knowledge accumulation and retention. One combination that stands out consistently is $\text{init}_{EMA}$ with $\text{deploy}_{EMA}$. This supports the findings of Stojanovski et al. (2022); Roth et al. (2024b) on small-scale continual learning and pretraining.

As the application of EMA model merging achieves a notably better balance between knowledge accumulation and retention than other methods, we call this approach *Best-in*-TIME. In the next section, we will explore the robustness of this strategy across different merging techniques. Please refer to Appendix E for additional EMA experiments.

### 3.3    WHAT IS THE BEST MERGE FOR BEST-IN-TIME?

Having identified the optimal initialization and deployment merging strategy, we now investigate the robustness of our finding by sweeping over merging techniques. In particular, we test 7 different merging techniques while keeping the *Best-in*-TIME initialization and deployment strategy. From Fig. 4, it is clear that all merging techniques perform very similarly. This indicates that, over a sufficiently long time horizon, all techniques converge to a similar behavior, echoing our results in Sec. 3.1. However, we do notice higher variance in the retention metric ($\mathcal{A}_{ZS}$).

### 3.4    SCALING UP TEMPORAL MODEL MERGING

We next scale temporal model merging up across three-axes: *model size*, *compute budget*, and *number of tasks* (results in Fig. 5 and Appendix E.2). All our experiments use the *Best-in*-TIME setup described previously, conducting hyperparameter-optimal EMA at each task.

**Scaling the Model.**    As we increase the model scale from $S/16$ (62.3M parameters) to $B/16$ (149.6M), $L/14$ (427.6M), and finally $g/14$ (1.37B) in Fig. 5 (left), we study the tradeoff between knowledge accumulation and retention over time. We compare sequential fine-tuning (circles) and *Best-in*-TIME (squares). *Best-in*-TIME scales well with model size, with larger models exhibiting increased affinity to merges over time. This extends and further corroborates offline merging

insights by Yadav et al. (2023), who showed that model scale facilitates merging. Moreover, while Roth et al. (2024b) and Ibrahim et al. (2024) highlight better continual fine-tuning with scale, we show temporal model merging to be substantially more effective across scale. For larger models all the way to the largest ViT-g/14, *Best-in*-TIME vastly outperforms or matches sequential fine-tuning and the multitask target in knowledge retention and positive backward transfer. Furthermore, scale facilitates equivalent degrees of knowledge accumulation between sequential fine-tuning and temporal model merging. Therefore, our model scaling results strongly suggest the use of temporal model merging solutions over standard continual fine-tuning methods.

**Scaling the Compute.** Keeping the underlying base model fixed to ViT-B/16, we next change the available compute budget by increasing the number of update steps per task. We compare a multitask model, trained on all tasks simultaneously, to a budget-optimal *Best-in*-TIME. The only hyperparameter for *Best-in*-TIME is the interpolation weight $w$. For each compute budget, there is a clear optimal choice of that hyperparameter (suboptimal runs shown as gray dots in Fig. 5 (right)). Higher values of $w$ put greater emphasis on accumulation, allowing optimal accumulation-retention trade-offs to be reached at lower compute budgets. However, for a larger compute budget, less aggressive temporal model merging can achieve higher absolute trade-offs. Note that in Fig. 5 (right), we report the geometric mean between accumulation and retention, corresponding to the right-most panel in previous plots. *Best-in*-TIME scales very well across compute budgets, *clearly approaching the multitask upper bound* in accumulation-retention balance at larger compute budgets.

**Scaling the Number of Tasks.** Given that all our results until now have been with $T{=}20$, we next study how *Best-in*-TIME performs as we increase the number of merging time-steps to much longer time-sequences: $T{=}50$ and $T{=}100$. *Best-in*-TIME remains the optimal method of choice across different initialization and deployment strategies. Please refer to Appendix E.2 for details.

## 4    CONCLUSION

In this work, we study *temporal model merging*, addressing the challenge of continually merging multimodal models as new tasks and data arrive, and new expert models are trained in succession. To formalize this setting, we propose TIME, a novel unifying framework breaking down temporal model merging into three key axes: (1) initialization phase defining starting weights before each task, (2) deployment phase denoting post-training expert model aggregation, and (3) the choice of merging technique. Using TIME, we conduct a large-scale systematic study uncovering crucial practical guidelines for temporal model merging. Our experiments on the FoMo-in-Flux benchmark spanning 63 datasets, showcase that accounting

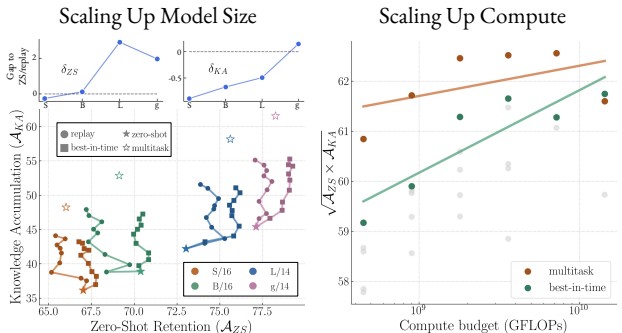

Figure 5: **Scaling up model merging.** *(left)* With scale, we observe continued improvements of model merging compared to the standard replay baseline. *(right)* Our *Best-in*-TIME method continues to improve with scaled total compute budget moving close to the multitask upper-bound. Gray points in the plot visualize suboptimal *Best-in*-TIME hyperparameter-instantiations.

for the temporal aspect is crucial, with standard offline merging techniques falling short in this dynamic setting. Moreover, we find the particular choice of merging technique matters far less than the merging strategy for initialization and deployment. Finally, we introduce *Best-in*-TIME, which scales favorably with model size and outperforms existing methods for continual multimodal pretraining. Our work provides a systematic entry point into temporal model merging and establishes best practices for this emerging field.

**Acknowledgements.** The authors would like to thank (in random order) Shyamgopal Karthik, Shashwat Goel, Ankit Sonthalia, Olivier Hénaff, Alexandre Ramé, and Daniel Marczak for helpful feedback. The plotting style in our work is inspired by figures from Beyer et al. (2022). The style of Fig. 2 is inspired by Figure 1 of Karamcheti et al. (2024). VU, KR, and SD thank the International Max Planck Research School for Intelligent Systems (IMPRS-IS). VU, KR, and SD also thank the European Laboratory for Learning and Intelligent Systems (ELLIS) PhD program for support. VU is supported by a Google PhD Fellowship in Machine Intelligence. SA is supported by a Newton Trust Grant. AP is supported by the Federal Ministry of Education and Research (BMBF), FKZ: 011524085B. AP and MB acknowledges financial support via the Open Philanthropy Foundation funded by the Good Ventures Foundation. MB is a member of the Machine Learning Cluster of Excellence, funded by the Deutsche Forschungsgemeinschaft (DFG, German Research Foundation) under Germany's Excellence Strategy – EXC number 2064/1 – Project number 390727645. ZA acknowledges the support from the German Research Foundation (DFG): SFB 1233, Robust Vision: Inference Principles and Neural Mechanisms, project number: 276693517 and ERC Grant DEXIM, project number: 853489. This research utilized compute resources at the Tübingen Machine Learning Cloud, DFG FKZ INST 37/1057-1 FUGG.

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

## A    Related Works

**Model Merging.** We provide a short overview of the model merging literature, detailed in these excellent surveys (Yadav et al., 2024a; Yang et al., 2024a). While both model aggregation through distillation Roth et al. (2024a); Cideron et al. (2024) and averaging checkpoints during training (Kaddour, 2022; Sanyal et al., 2024; Li et al., 2024) have shown success, the requirement of additional compute limits practicability of these methods (Prabhu et al., 2023b). Instead, recent work (Wortsman et al., 2022b;a; Ilharco et al., 2022; 2023; Rame et al., 2023; Sanyal et al., 2024; Sung et al., 2023; Pari et al., 2024; Nylund et al., 2023; Zaman et al., 2023; Stoica et al., 2024; Wang et al., 2024b; He et al., 2024; Oh et al., 2024; Shen et al., 2024; Sharma et al., 2024; Goddard et al., 2024; Yadav et al., 2024a; Xiong et al., 2024; Yang et al., 2024b; Lu et al., 2024; Zheng & Wang, 2024; Nasery et al., 2024) has shown the effectiveness of training-free weight averaging and interpolation of fine-tuned expert models to produce an improved base model, benefiting from (linear) mode connectivity in models fine-tuned from a single pre-trained checkpoint Izmailov et al. (2018); Ramé et al. (2024); Neyshabur et al. (2020); Frankle et al. (2020); Ainsworth et al. (2023). These insights have been extended into weight-averaged reward models Ramé et al. (2024), policy models Ramé et al. (2024) with spherical interpolation, and KL-constrained RLHF Lin et al. (2024); Liu et al. (2024); Munos et al. (2024); Gorbatovski et al. (2024). Works such as Fisher-Merge Matena & Raffel (2022), TIES Yadav et al. (2023), RegMean Jin et al. (2023), MATS Tam et al. (2024a), DELLA Deep et al. (2024), DARE Yu et al. (2024), Breadcrumbs Davari & Belilovsky (2025), evolutionary merging Akiba et al. (2024) and MagMax Marczak et al. (2024) have explored merging strategies beyond simple interpolation to determine which weights should be merged across expert models. These methods have different benefits for in- and out-of-distribution generalization across domains Tam et al. (2024b), though recently they have been shown to perform similarly at scale Yadav et al. (2024b). Additionally, some works have explored the initialization dimension for effectively merging models (Choshen et al., 2022; Don-Yehiya et al., 2022; Zhou et al., 2024; Marczak et al., 2024). In this work, we propose a unifying framework for temporal merging and conduct the most comprehensive study of this topic to date.

**Continual Pretraining** extends beyond standard Continual Learning (Prabhu et al., 2023a; Roth et al., 2024b), focusing on large-scale model updates starting from pretrained foundation models Ibrahim et al. (2024); Garg et al. (2024); Roth et al. (2024b); Gui et al. (2024); Prabhu et al. (2023c) and addressing more complex and substantial update tasks Lin et al. (2021); Cai et al. (2021); Liska et al. (2022); Garg et al. (2024); Bornschein et al. (2023); Roth et al. (2024b). There has been limited exploration into using model merging for continual pretraining (Marczak et al., 2024; Alexandrov et al., 2024; Stojanovski et al., 2022; Roth et al., 2024b), as most prior works focus on training strategies including regularization objectives and learning-rate schedules (Roth et al., 2024b; Prabhu et al., 2023b; Garg et al., 2024; Ibrahim et al., 2024; Srivastava et al., 2024; Li et al.; Yıldız et al., 2024; Thede et al., 2024; Ostapenko et al., 2022; Mendieta et al., 2023). We keep the training strategy fixed, and provide an in-depth exploration beyond simple merging techniques.

## B    Details of the merging methods

Denoting the number of models to merge at timestep $t$ as $M_t$ (with $t = 0$ and $M_t = M$ for standard model merging), we can define these methods as follows:

**Weight Averaging Regent's et al. (1996); Wortsman et al. (2022b); Ilharco et al. (2022); Stojanovski et al. (2022); Roth et al. (2024b)** simply employs a uniformly weighted, element-wise average over all models $\theta_{t,i}$, resulting in a merge function $f_{\text{merge}}^{\text{WA}}$:

$$\theta_t = \frac{1}{M_t} \sum_i \theta_{t,i}. \tag{2}$$

**SLERP Shoemake (1985); Ramé et al. (2024)** assumes weights to live on a hypersphere, and consequently conducts interpolation along a curved path connecting weight entries. In particular, for two models $\theta_{t,1}$ and $\theta_{t,2}$ deriving from some base weight $\theta_{t-1}$ and the corresponding task vectors

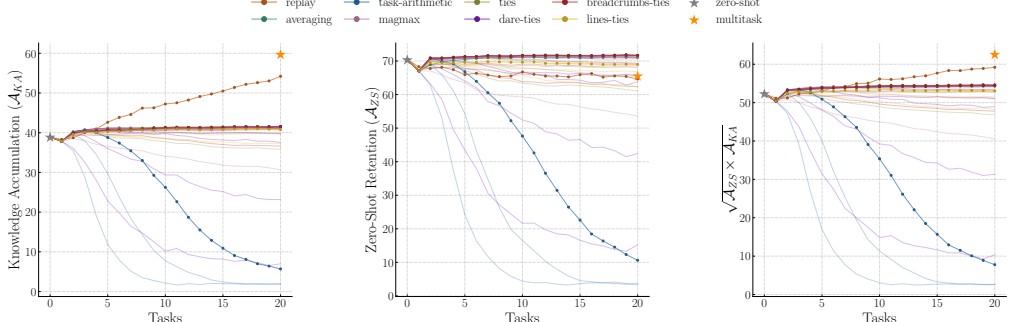

Figure 6: **Offline merging methods struggle with TIME.** All tested merging techniques perform extremely poorly, and are unable to adapt to the temporal setting, underperforming even a simple replay baseline that sequentially trains the base model on task-replayed data.

$\delta_{t,i} = \theta_{t,i} - \theta_{t-1}$, SLERP with interrpolation weight $\lambda$ is defined as

$$\theta_t = \theta_{t-1} + \frac{\sin(1-\lambda)\Omega_{1,2}}{\sin \Omega_{1,2}} \cdot \delta_{t,1} + \frac{\sin \lambda \Omega_{1,2}}{\sin \Omega_{1,2}} \cdot \delta_{t,2} \tag{3}$$

with $\Omega_{1,2}$ being the angle between task vectors $\delta_{t,1}$ and $\delta_{t,2}$. We denote the corresponding merge function $f_{\text{merge}}^{\text{SLERP}}$.

**Task Arithmetic Ilharco et al. (2023)** defines the merge as a function over task vectors $\delta_{t,i} = \theta_{t,i} - \theta_{t-1}$ for each weight $\theta_{t,i}$ fine-tuned from $\theta_{t-1}$. This introduces a simple merge formalism $f_{\text{merge}}^{\text{TA}}$ for weighted parameter averaging with a scale $\lambda$:

$$\theta_t = \theta_{t-1} + \lambda \frac{1}{M_t} \sum_i \delta_{t,i} \tag{4}$$

**TIES Yadav et al. (2023)** builds on the task arithmetic formalism through controlled pruning of task vector entries with low magnitude. Moreover, the sign for each final merged parameter is set based on the sign of the highest total magnitude across the merge candidates. The final update follows basic task arithmetic, only for entries with matching signs. We refer to the respective merge function as $f_{\text{merge}}^{\text{TIES}}$.

**DARE Yu et al. (2024)** is a similar extension of task arithmetic, but instead of targetted pruning, it randomly zeroes out task vector entries using a random mask $Z_i \sim \text{Bernoulli}(p)$ and masking probability $p$. Final task vector values for $f_{\text{merge}}^{\text{DARE}}$ are then rescaled based on $p$:

$$\delta_{t,i}^{\text{DARE}} = \frac{(1-Z_i)\delta_{t,i}}{1-p}. \tag{5}$$

**Model Stock Jang et al. (2024)** provides a geometric extension of simple weight averaging as done in Model Soup Wortsman et al. (2022a) by incorporating base weights $\theta_{t-1}$ into the merging process. Given fine-tuned weights $\theta_{t,1}$ and $\theta_{t,2}$, the Model Stock merge $f_{\text{merge}}^{\text{Stock}}$ is defined as follows:

$$\theta_t = \frac{2 \cdot \cos \Omega_{1,2}}{1 + \Omega_{1,2}} \cdot (\theta_{t,2} - \theta_{t,1}) + \left(1 - \frac{2 \cdot \cos \Omega_{1,2}}{1 + \cos \Omega_{1,2}}\right), \tag{6}$$

utilizing angle $\Omega_{1,2}$ between task vectors $\delta_{t,1}$ and $\delta_{t,2}$.

**Breadcrumbs Davari & Belilovsky (2025)** deploys another variation on task arithmetic for model merging. In particular, for a given task vector $\delta_{t,i}$, extreme left and right tails of the absolute magnitude distribution in $\delta_{t,i}$ are zeroed out with left and right thresholds $\beta$ and $\gamma$. The modified task vectors $\delta_{t,i}^{\text{Bread}}$ are then applied on base weights $\theta_{t-1}$ following the task arithmetic setup, and giving $f_{\text{merge}}^{\text{Bread}}$.

**MagMax Marczak et al. (2024)** also uses task vectors—given multiple task vectors $\delta_{t,i}$ (with increments possible along both time $t$ and count axis $i$), the final task vector $\delta_t$ is yielded through maximum magnitude entry selection; copying the largest magnitude entries across all $\{\delta_{t,i}\}$ into $\delta_t$, giving $f_{\text{merge}}^{\text{Max}}$.

**LiNeS Wang et al. (2024a)**, for Liayer-increasing Network Scaling, scales weight updates based on their respective layer depth enabling early layers to remain close to original pretraining weights (cf. Neyshabur et al. (2020)). Given task vectors $\delta_{t,i}$, now broken down across model layers $\delta_{t,i}^l$ with $l \in [1, ..., L]$ and $L$ the number of layers, LiNeS follows the base task arithmetic merging formalism, but updates task vectors as

$$\delta_{t,i}^{\text{LiNeS}} = \text{concat}\left(\lambda^{l=1}\delta_{t,i}^{l=1}, ..., \lambda^{l=L}\delta_{t,i}^{l=L}\right) \tag{7}$$

with layer-scaled interpolation weights $\lambda^l = \alpha + \beta\frac{l-1}{L-1}$ and hyperparameters $\alpha, \beta$, giving $f_{\text{merge}}^{\text{Lines}}$.

## C  PLOTTING STYLE

Across TIME, we utilize a common plotting style to visualize our results—with three base subplots (see for *e.g.*, Fig. 5):

- Knowledge Accumulation ($\mathcal{A}_{KA}$) versus number of tasks over time. In this plot, a gray star indicates the base-weight zero-shot performance on adaptation datasets. An orange star indicates an upper bound achieved through jointly training on all the data at once, with no separation over time.

- Zero-Shot Retention ($\mathcal{A}_{ZS}$) versus number of tasks over time. Similar to $\mathcal{A}_{KA}$ versus tasks, this plot visualizes merging results for TIME-variants, but measuring performance on withheld evaluation datasets. Again, gray and orange star indicate base and joint training lower and upper bounds, respectively.

- Finally, we also aggregate both previous plots into one showcasing the progression of merged performance geometric mean $\sqrt{\mathcal{A}_{ZS} \times \mathcal{A}_{KA}}$ over time; utilizing the same star indication as in the previous subplots.

The only deviation from this plotting style is Fig. 5. The left panel visualizes the trajectory across tasks in the $\mathcal{A}_{KA}$ - $\mathcal{A}_{ZS}$ space. Here, full-colored stars reference base model performance and hollow stars the corresponding joint training upper bounds. The right panel shows the geometric mean of $\mathcal{A}_{KA}$ and $\mathcal{A}_{ZS}$ at the end of the last task for different compute budgets.

Finally, several plots such as Figs. 4, 6 and 7 show the extensive scale of our experiments through background visualizations of sub-optimal hyperparameter choices in lighter colors (as opposed to the optimal choices using darker coloring). This plotting style is loosely inspired by Beyer et al. (2022).

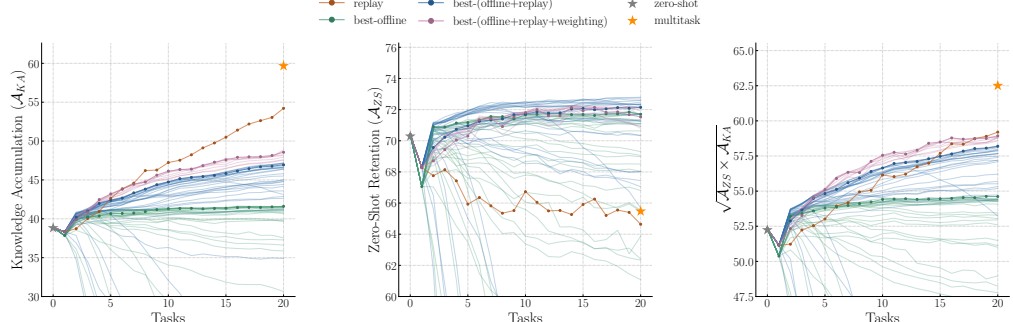

Figure 7: **Improving *offline* merging.** We identify two simple methods for adapting offline-merging methods to the temporal setting: (1) replaying data from previous tasks (best-(offline+replay)) and (2) recency-biased weighting of task checkpoints (best-(offline+replay+weighting)). With these method improvements, offline merging methods can match the replay baseline.

# D    ADDITIONAL OFFLINE MERGING EXPERIMENTS

## D.1    REPLAY AND TIME-WEIGHTING

In this section, we analyze extensions to offline methods that can help close the gap to the replay baseline. As the continual fine-tuning baseline *replays* on past data from all previous tasks while training at the current task $t$, can this task data-mixing also help offline merging methods?

**Data replaying improves offline merging.** Since offline methods operate entirely under a task-independent assumption, they fail to capture any temporal dependencies. Fig. 7 shows that simply applying data-replay on top of standard offline merging leads to significant boosts in the overall performance. For instance, best-(offline+replay) achieves $58.2\%$ compared to best-offline at $54.6\%$, bringing it closer to the replay baseline. However, a notable performance gap remains, with best-(offline+replay) at $58.2\%$ falling short of replay at $59.1\%$.

**Recency-biased weighting helps.** Next, unlike in standard *offline* averaging, where all task checkpoints are weighted uniformly, we impose temporal ordering via non-uniform weighting for offline merging. We explore several recency-biased, non-uniform weighting schemes, assigning higher weights to more recent tasks to account for the temporal nature of the setting.

We explore various discounting schemes: logarithmic, quadratic, exponential, and cubic, applied to the best offline merge replay method from the previous experiment (please refer to the supplementary for details). As shown in Fig. 7, these schemes improve performance, with best-(offline+replay+weighting) reaching $58.9\%$, yet still falling slightly short of the replay baseline at $59.1\%$. These results provide strong evidence that accounting for the new temporal axis is crucial for effective temporal model merging, even when implemented as an extension of offline merging. **Key takeaway:** accounting for the time aspect is crucial for effective temporal model merging, even as an extension on top of standard offline merging. Still, a small gap to the simple replay baseline remains.

## D.2    REVERSED NON-UNIFORM WEIGHTING SCHEMES

In Fig. 8, we found that a simple yet effective method for boosting the performance of offline merging methods is recency-biased non-uniform weighting, i.e. giving larger weights to more recent checkpoints while merging. Here, we ask the question—what if we reversed the weighting schemes such that we give larger weights to older task checkpoints? From Fig. 8, we indeed observe that such a reverse strategy performs worse than the best recency-biased weighting schemes, since the knowledge accumulation ability is hampered by giving more emphasis to older tasks. However, note that such a sub-optimal reverse weighting strategy is still better than the pure offline merging strategy with *no replay*. This helps further ablate the exact importance of *replay* and *non-uniform weighting* for improving pure offline-merging techniques in the presence of the time axis.

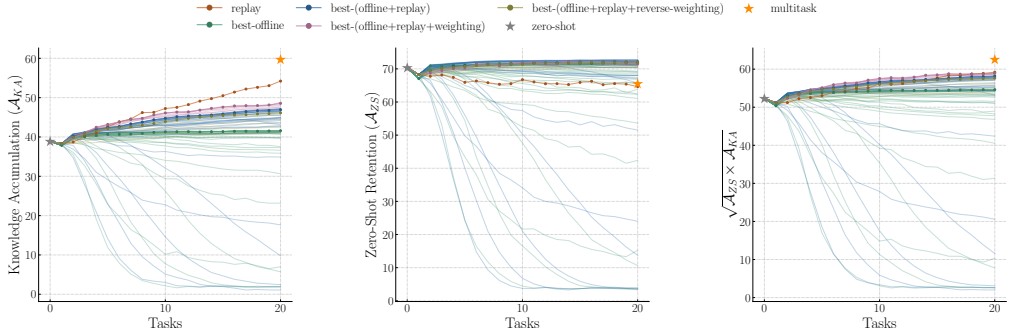

Figure 8: **Effect of reverse-weighting for offline merging techniques.** We find that reversing the weighting scheme that yielded consistent boosts from Fig. 7 is sub-optimal—indeed, it performs worse than the offline merging with replay methods.

# E  ADDITIONAL EMA EXPERIMENTS

## E.1  TASKS AS DATASETS

In the main text, we presented all results using a data stream that randomly mixes concepts from different datasets into a coherent set of tasks—following the *random* data-stream in Roth et al. (2024b). Here, we relax this constraint and re-run our experiments using individual datasets as tasks, consistent with the standard model merging literature (Ilharco et al., 2022; 2023; Yadav et al., 2023). Specifically, we use the *dataset-incremental* stream from Roth et al. (2024b). Even in this setup, we reproduce our main findings. In Fig. 9, we confirm the results from Fig. 6, showing that all offline merging techniques perform poorly when exposed to the axis of time, failing to even match the performance of a simple continual fine-tuning *replay* baseline. Additionally, in Fig. 10, we corroborate the results from Fig. 3, demonstrating that the *best-in-*TIME method remains the most effective temporal model merging approach. We also confirm that the choice of model merging technique is far less critical for temporal model merging than the initialization and deployment strategies.

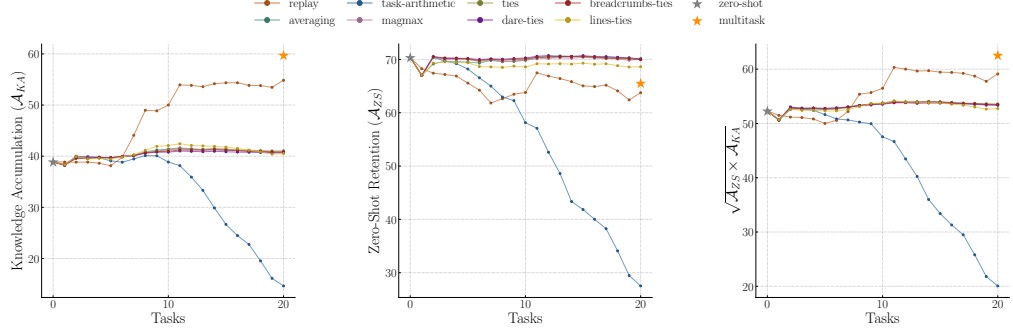

Figure 9: **Offline merging techniques still struggle in the tasks-as-datasets setting.** Switching from the *random* data-stream (Fig. 6 in the main paper) to the *dataset-incremental* stream, which aligns more closely with the standard multi-task merging literature setups, reveals that offline merging techniques still severely underperform compared to the simple *replay* baseline.

## E.2  LONGER TASK SEQUENCES

To test the robustness of our findings in Sec. 3.2, we repeat the experiment shown in Fig. 3 on a longer sequence with the number of tasks $T = 50$ (Fig. 11). For 50 tasks, *Best-in-*TIME still strikes the optimal balance between knowledge accumulation and zero-shot retention. One notable difference with respect to Fig. 3 is the large initial advantage of the zero-shot initialization strategy combined with the EMA deployment strategy. When the learning horizon is further extended to 100

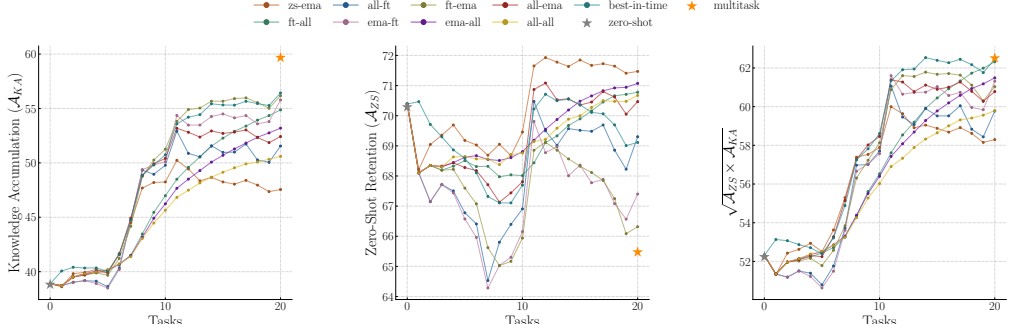

Figure 10: **Dataset-Incremental TIME Exploration.** We replicate the results from Fig. 3 using the dataset-incremental stream instead of the random stream. The main takeaways remain unchanged: initialization and deployment strategies primarily determine temporal merging performance, and the EMA-averaging initialization and deployment strategy utilized in *Best-in*-TIME is the best approach.

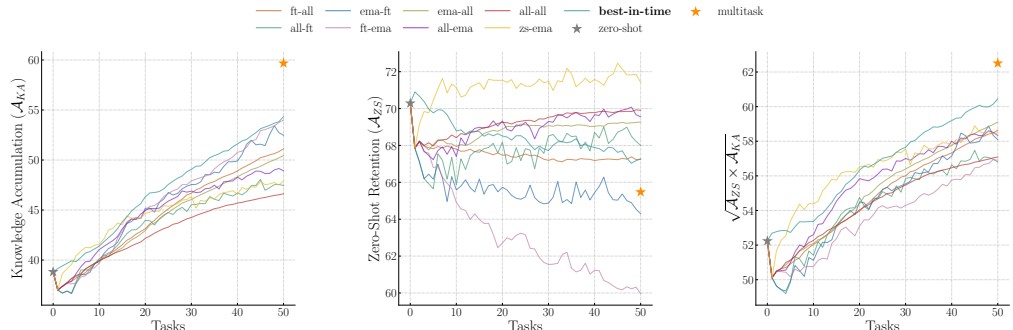

Figure 11: **A long journey through TIME.** We compare all valid combinations of initialization and deployment protocols on a longer sequence of 50 tasks. *Best-in*-TIME remains the best in balancing knowledge accumulation and zero-shot retention.

tasks, this initial advantage is maintained, establishing the zero-shot initialization approach as the best-performing method, as shown in Fig. 12. Although the double EMA variant surpasses zero-shot initialization in knowledge accumulation, its poor retention relegates it to third place on the combined metric. In this exploration we re-use the optimal interpolation weight from the 20 task scenario, which may no longer be ideal for longer horizons, as it directly influences the balance between knowledge accumulation and zero-shot retention.

### E.3    VARIANCE ANALYSIS ACROSS RUNS

To put our results from Sec. 3.3 in perspective, we quantify the variance across runs for a single merging method. Specifically, we run *Best-in*-TIME three times and show the mean and standard deviation across runs in Fig. 13. Comparing this to Fig. 4 reveals that the best results for different methods fall within the standard deviation of multiple runs of the same method. In particular, for the last task, the standard deviation of the geometric mean of knowledge accumulation and zero-shot retention is 0.96.

## F    HYPERPARAMETER DETAILS

In an effort to remove any confounding factors, we conduct an extensive hyperparameter sweep, to the best of our abilities, for each individual merging technique for Figs. 4, 6 and 7. We list the hyperparameter ranges swept over for each technique below:

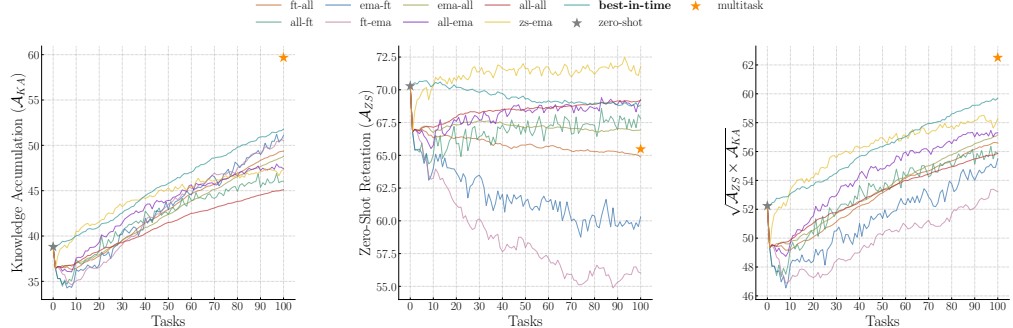

Figure 12: **An even longer journey through TIME.** We compare all valid combinations of initialization and deployment protocols on a longer sequence of 100 tasks. *Best-in*-TIME still remains the best approach balancing knowledge accumulation and retention, measured as the geometric mean of the two metrics in the right-most figure.

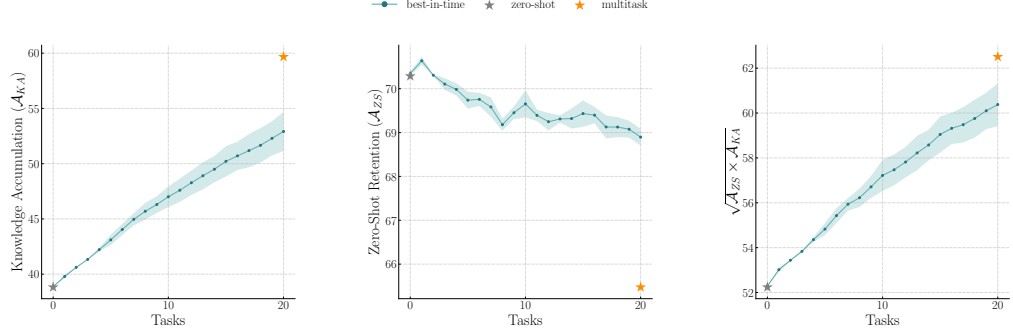

Figure 13: The mean and standard deviation across three runs of *Best-in*-TIME.

- **Weight Averaging.** For the offline merging, we use a standard merging coefficient of $\frac{1}{N}$, where $N$ is the number of task checkpoints to merge.
- **SLERP.** In SLERP, as we can only merge two checkpoints at a time, we sweep over the following weight-coefficients: $\{0.1, 0.3, 0.5, 0.7, 0.9\}$.
- **Task-Arithmetic.** We sweep over the scaling factor: $\{0.1, 0.2, 0.3, 0.4, 0.5, 0.6, 0.7, 0.8, 0.9, 1.0\}$
- **TIES.** We sweep over the scaling factor: $\{0.1, 0.2, 0.3, 0.4, 0.5, 0.6, 0.7, 0.8, 0.9, 1.0\}$ and the pruning-fraction: $\{0.1, 0.2, 0.3, 0.4, 0.5, 0.6, 0.7, 0.8, 0.9, 1.0\}$.
- **DARE-TIES.** We sweep over the scaling factor: $\{0.1, 0.2, 0.3, 0.4, 0.5, 0.6, 0.7, 0.8, 0.9, 1.0\}$ and the pruning-fraction: $\{0.1, 0.2, 0.3, 0.4, 0.5, 0.6, 0.7, 0.8, 0.9, 1.0\}$.
- **Breadcrumbs-TIES.** We sweep over the scaling factor: $\{0.1, 0.2, 0.3, 0.4, 0.5, 0.6, 0.7, 0.8, 0.9, 1.0\}$ and the pruning-fraction: $\{0.1, 0.2, 0.3, 0.4, 0.5, 0.6, 0.7, 0.8, 0.9, 1.0\}$.
- **MagMax.** We sweep over the scaling factor: $\{0.2, 0.4, 0.8, 1.0\}$.
- **LiNeS-TIES.** We keep $\alpha$ fixed to 0.5, and sweep $\beta$: $\{0.2, 0.5, 0.8\}$ and prune-fraction: $\{0.2, 0.5, 0.8\}$ as recommended in the original paper (Wang et al., 2024a).

