# OpenReview forum: "How to Merge Multimodal Models Over Time?"
_ICLR.cc/2025/Workshop/MCDC — MCDC @ ICLR 2025_

### Official Review · Reviewer_9Py6 · 2025-02-27

**Rating:** 8
**Confidence:** 4
**Fit:** 5

**Summary:**

The authors propose a framework TIMES under which they study temporal merging for continual learning of tasks. They identify three axes a) Initialization b) deployment c) merging technique and study existing approaches for each of them.

**Reason For Giving A Higher Score:**

Experiments presented in the paper and insights drawn from it are helpful to the community

**Reason For Giving A Lower Score:**

None

**Strengths And Weaknesses:**

-Rigorous evaluation with multiple merging methods

- Findings in the paper about merging methods not mattering much. Additionally, initialization and deployment seems to matter more is insightful

**Suggestions:**

- If time permits, I suggest experiment with https://arxiv.org/pdf/2312.04339. Here they have a method to merge models using a conjugate gradient algorithm and combine initialization from different merging methods and improve upon them.

- It is weird that multitask training is reducing the zero-shot performance of the model in Figure 3. Can you explain this behavior?
- ( any_init, deployEMA, f_merge) is kind of misleading since there is no EMA. Equation 1 does merging of previous model with current expert model with f_merge method. Where does the exponential moving average part come in?

---

### Official Review · Reviewer_aajb · 2025-02-28

**Rating:** 9
**Confidence:** 4
**Fit:** 5

**Summary:**

The submission proposes TIME (Temporal Integration of Model Expertise), a unifying framework for temporal model merging—that is, merging multiple specialist or _expert_ models over time as new tasks arrive and expert checkpoints are produced. The key premise is that standard _offline_ model merging, which merges experts only once (after all tasks), fails in the more realistic continual or temporal scenario. TIME systematically explores three primary design decisions (initialization, deployment, and merging technique) that shape how each new expert is created and eventually merged to form a “global” model.

The paper’s experiments, conducted on the FoMo-in-Flux benchmark (63 multimodal datasets) and tested on a CLIP-based model architecture, demonstrate the following:
- The _temporal_ dimension is critical. Standard offline merging underperforms in sequential scenarios.
- Sophisticated merging methods (like TIES, Task Arithmetic, etc.) offer only marginal gains over simple weighted averaging when used over many time steps.
- The choice of initialization (e.g., reusing an exponential moving average of previous experts) and deployment (e.g., also using an EMA across trained checkpoints) is more critical than the minor differences among merging algorithms.
- Temporal model merging scales favorably with larger models, more compute, and longer task sequences, often outperforming naive fine-tuning or offline baselines.

This work fills a gap in the literature by showing how to merge multiple models in a long-running, evolving setting, clarifying best practices (especially around initialization/deployment with EMA) and identifying open challenges (e.g., memory constraints or highly divergent tasks).

**Reason For Giving A Higher Score:**

1. **Novel Problem Setting**: This is one of the first large-scale, systematic explorations of temporal model merging, bridging offline merging methods with continual learning challenges.
2. **Practical Significance**: The paper yields actionable recommendations (particularly around EMA for initialization/deployment) that are likely to benefit real-world practitioners who retrain or continually update large models.
3. **Comprehensive Empirical Evidence**: The experiments are extensive, scaling across dimensions (model size, compute, # tasks), which is relatively rare in this domain.

**Reason For Giving A Lower Score:**

1. **Limited Theory**. Readers looking for deeper theoretical grounding or interpretability of weight merges (beyond empirical success) might find the paper lacking.
2. **Memory Constraint Oversight**. Storing every expert could be infeasible for certain real-world or extremely long-horizon applications, and the paper mainly sidesteps that issue.

**Strengths And Weaknesses:**

### Strengths

- **Well-Structured Framework**. The paper offers a clear taxonomy via TIME, systematically categorizing initialization, deployment, and merging technique. This provides a modular approach that can be adapted to other model families and tasks.
- **Thorough Empirical Study**. Multiple large-scale experiments on a multimodal continual pretraining benchmark (FoMo-in-Flux) bolster the paper’s conclusions. The authors carefully compare different strategies and systematically show the trade-offs.
- **Useful Practitioner Takeaways**. The finding that simpler merges work well, combined with the strong effect of exponential moving average initialization/deployment, is both practical and broadly relevant for real-world continual learning scenarios.
- **Scalability Experiments**. The paper investigates model size, number of tasks, and compute budget, all of which are highly relevant for modern large-scale (foundation) models.

### Weaknesses

- **Memory/Storage Constraints**. The presented method retains a buffer of all trained checkpoints (expert models). For extremely long task sequences or very large models, storing hundreds of checkpoints is impractical. The text mentions this concern but does not propose advanced checkpoint selection or compression methods.
- **Limited Theoretical Insight**. Although the experiments are comprehensive, the paper is relatively light on rigorous theoretical justifications for why EMA merges so consistently dominate other merging strategies. A deeper discussion or analysis of underlying geometry or mode connectivity would enrich the results.

**Suggestions:**

- To address memory growth, experiment with either pruning older experts or storing only partial snapshots. This would practically demonstrate how TIME might scale in extremely long-horizon settings.
- If possible, add an outline or short theoretical argument on how EMA merges preserve “linear mode connectivity” (or some variant) across tasks. This would lend more insight into why it consistently outperforms other merges.
- Provide (even briefly) a per-task breakdown of performance. While the authors do show aggregated metrics, seeing how each new task’s knowledge is integrated might further clarify merging trade-offs.
- In Section 2's first paragraph (Notation), it seems a bit unclear what would the variable $t$ represent: it seems to be simultaneously used for tasks and for time steps. It would be nice to reword both notations.

---

### Official Review · Reviewer_4tfY · 2025-03-01

**Rating:** 7
**Confidence:** 5
**Fit:** 4

**Summary:**

In summary, this is a good paper in the area of model merging. The paper introduces a novel framework, TIME (Temporal Integration of Model Expertise), which addresses the understudied problem of temporal (continual) model merging.

**Reason For Giving A Higher Score:**

refer to strengths and weaknesses.

**Reason For Giving A Lower Score:**

refer to strengths and weaknesses.

**Strengths And Weaknesses:**

Strength:

1.The paper introduces a novel framework, TIME (Temporal Integration of Model Expertise), which addresses the understudied problem of temporal model merging.
2. The authors break down temporal model merging into three key axes—initialization, deployment, and merging techniques—providing a structured approach to understanding and implementing temporal merging. This framework is well-defined and allows for a systematic exploration of the design space.

Weaknesses:

1. While the paper provides extensive empirical results, it lacks a theoretical analysis of why certain initialization and deployment strategies work better than others. A deeper theoretical understanding could strengthen the paper and provide more generalizable insights.
2. The paper concludes that complex merging techniques provide marginal benefits. I don't think that's entirely true. For larger models, simple model merging approaches can work well and complex merging techniques show marginal benefits, but for smaller models, complex approaches can sometimes have significant performance advantages. Therefore, the effectiveness of merging techniques can vary depending on the model size.
3. I suggest the authors consider citing the recent concurrent  work by Anke Tang et al., "Merging Models on the Fly Without Retraining: A Sequential Approach to Scalable Continual Model Merging," as it provides complementary insights into continual model merging and could further enrich the discussion on temporal merging strategies.

**Suggestions:**

Refer to the weaknesses.

---

### Decision · Program_Chairs · 2025-03-06

**Decision:**

Accept

**Comment:**

This paper has been highly appreciated by all reviewers. We recommend taking suggestions into consideration for the final version of the manuscript.